# Genetics in Cartilage Lesions: Basic Science and Therapy Approaches

**DOI:** 10.3390/ijms21155430

**Published:** 2020-07-30

**Authors:** Dawid Szwedowski, Joanna Szczepanek, Łukasz Paczesny, Przemysław Pękała, Jan Zabrzyński, Jacek Kruczyński

**Affiliations:** 1Orthopedic Arthroscopic Surgery International (O.A.S.I.) Bioresearch Foundation, Gobbi N.P.O., 20133 Milan, Italy; dszwedow@yahoo.com; 2Department of Orthopaedics and Trauma Surgery, Provincial Polyclinical Hospital, 87100 Torun, Poland; 3Centre for Modern Interdisciplinary Technologies, Nicolaus Copernicus University, 87100 Torun, Poland; 4Orvit Clinic, Citomed Healthcare Center, 87100 Torun, Poland; drpaczesny@gmail.com (Ł.P.); zabrzynski@gmail.com (J.Z.); 5Faculty of Medicine and Health Sciences, Andrzej Frycz Modrzewski Krakow University, 30705 Krakow, Poland; pekala.pa@gmail.com; 6Department of General Orthopaedics, Musculoskeletal Oncology and Trauma Surgery, Poznan University of Medical Sciences, 60512 Poznań, Poland; jacek@man.poznan.pl

**Keywords:** cartilage pathology, osteoarthritis, gene therapy, chondral lesions

## Abstract

Cartilage lesions have a multifactorial nature, and genetic factors are their strongest determinants. As biochemical and genetic studies have dramatically progressed over the past decade, the molecular basis of cartilage pathologies has become clearer. Several homeostasis abnormalities within cartilaginous tissue have been found, including various structural changes, differential gene expression patterns, as well as altered epigenetic regulation. However, the efficient treatment of cartilage pathologies represents a substantial challenge. Understanding the complex genetic background pertaining to cartilage pathologies is useful primarily in the context of seeking new pathways leading to disease progression as well as in developing new targeted therapies. A technology utilizing gene transfer to deliver therapeutic genes to the site of injury is quickly becoming an emerging approach in cartilage renewal. The goal of this work is to provide an overview of the genetic basis of chondral lesions and the different approaches of the most recent systems exploiting therapeutic gene transfer in cartilage repair. The integration of tissue engineering with viral gene vectors is a novel and active area of research. However, despite promising preclinical data, this therapeutic concept needs to be supported by the growing body of clinical trials.

## 1. Introduction

Articular cartilage is a highly specialized connective tissue. Its most important components include the extracellular matrix (ECM), comprising of chondrocytes, collagen fibers (mainly type II and IX), small non-collagen proteins (e.g., aggrecan, high molecular weight proteoglycan and cartilage oligomeric matrix protein (*COMP*)), a small volume of non-collagenous proteins, glycoproteins such as fibronectin and water [1]. This composition is regulated by chondrocytes in response to the changes in their chemical and mechanical environment. 

In addition to structural elements, chondrocytes produce substances that perform physiological functions: enzymes and cytokines. Such enzymes include metalloproteinases, adamalysins (A Disintegrin and Metalloproteinase Domain; *ADAMs*), serine and cysteine proteases, and their inhibitors [2,3]. Proinflammatory cytokines (e.g., *IL*-*1β, TNFα*, *IL*-*6, IL*-*15*, *IL*-*17*, and *IL*-*18*) participate in the turnover of the cartilaginous matrix, stimulating chondrocytes to increase the production of enzymes to enhance matrix degradation, while *IL*-*4*, *IL*-*10*, and *IL*-*13* inhibit this process [4,5]. Human articular chondrocytes express a constitutive complex of major histocompatibility system (MHC) class I, which are molecules that regulate complement activation. After their activation, such as under the influence of *IFN*-*α*, *IL*-*1*, *TNF*-*α,* or as a result of inflammatory joint diseases, chondrocytes also express MHC class II and *ICAM*-1 intercellular adhesion molecules [6]. Numerous studies have shown that chondrocytes also have tissue-specific antigens that induce the production of antibodies in patients with cartilage transplants, as well as in patients with rheumatoid arthritis (RA) and osteoarthritis (OA). The role of the genes coding structural components of cartilage and regulatory enzymes in the pathogenesis of osteochondral pathologies concerning inter- and intracellular signaling pathways is still under investigation.

Understanding and better characterizing the genetic basis of cartilage damage is important for several reasons. First, it can be useful to identify early and specific diagnostic biomarkers signaling the beginning of degradation; identifying such markers can play a crucial role in predicting and monitoring disease progression. As well, the identification of predisposing genes also aids in the stratification of risk groups and the introduction of preventative strategies. However, the most important consideration from a clinical perspective is the therapeutic advantages; the identification of genetic determinants is increasingly an introduction to the development of new individualized therapeutic protocols. Gene therapies are promising for diseases causing cartilage damage, and nanoconstructions containing low molecular RNA or vectors with gene expression modulators are being developed.

Biomechanical, genetic, inflammatory, and hormonal components are important for the development and progression of cartilage tissue diseases. The purpose of this review is to present the diversity of genetic changes in cartilage and the prospective use of gene therapy in joint pathologies.

## 2. Genetic Basis of Cartilage Pathology

Disorders of cartilage homeostasis are a significant feature of the pathological state of the tissue. In a state of imbalance between catabolic and anabolic processes, degeneration prevails over regeneration, resulting in the loss of cartilage matrix. 

An important cause of these cartilaginous pathologies is the successive changes at the genome and transcriptome levels, as well as their effects in the form of proteomic changes. The molecular markers associated with the development of cartilage pathologies can be identified using molecular biology techniques (single-gene analysis by polymerase chain reaction or genome-wide testing, such as with microarrays) or bioinformatics and computational biology methods (e.g., meta-analyzes, association studies) [7,8,9]. The advent of sequencing methodologies (next-generation sequencing or massively parallel high-throughput DNA sequencing) has provided new possibilities for searching, as well as quantitative and functional analysis of genome and transcriptome. The main purposes of large-scale genomic screens are searching for novel biomarkers and their genetic signatures, and these are also associated with function in physiological and pathological conditions [7,10,11,12]. Several scientific studies show a correlation between abnormalities in the behavior of cartilage homeostasis and structural changes within DNA (mainly single nucleotide polymorphisms type), differentiated gene expression, as well as epigenetic regulations (such as microRNA and promotor region methylation). Genome and transcriptome changes have been described for most cartilage diseases; however, they were best characterized for RA and OA. Genetic biomarkers of cartilage structure damage and metabolism disorders, with their therapeutic potential, are presented in Table 1. 

The genetic changes in cartilage are regulated directly and indirectly via genes associated with tissue metabolism. Quantitative and qualitative changes in key genes trigger a cascade of changes that lead to disorders in several signaling pathways (such as signal transduction, *NF*-*ĸB*, *JAK*/*STAT, Wnt,* and *mTOR*) [51]. The initiation and progression of osteoarthritis may be associated with, among others, changes in *IL-1β* expression levels that affect signal transduction through the *NF-ĸB* pathway [52]. Cartilage degradation by the proteasome–ubiquitin system and intra-cartilage ossification have been correlated with abnormalities in the Wnt pathway mediated by *Sox9* and *FRZB* [53,54]. In turn, genetic changes in *IL-1*, *IL-6,* and *TNFα* affect, via the *NF*-*ĸB* and *JAK/STAT* pathways, the induction of rheumatoid arthritis [55,56]. Therefore, the introduction of inhibitors of overexpressed transcription factors and proinflammatory cytokines may have clinical benefits in the regulation of chondrocyte proliferation and differentiation [36]. The activity, concentration, or expression of the above-mentioned molecules is relatively easy to determine (at the gene or protein level) in biological fluids such as blood, urine, and joint fluid. Markers of cartilage degeneration have a moderate or good correlation with clinical and radiological changes in the course of degenerative diseases, especially OA and RA [25].

Cartilage diseases are often accompanied by synovitis [57] (Figure 1). Symptoms of the inflammatory state are the proliferation of synoviocytes and tissue hypertrophy. Synoviocytes release inflammatory mediators and matrix-degenerating enzymes into the joint. Their activation occurs due to the action of inflammatory mediators and cartilage matrix molecules, initiating a feedback cycle within the synovium, which results in progressive degeneration of the joint. 

## 3. Diagnostic and Therapeutic Biomarkers

Metalloproteinases, inflammatory factors, signaling molecules, and transcription factors belong to the best-described groups of enzymes and their genes involved in the pathogenesis of cartilage tissue disease [36,58]. Genetic changes within these gene superfamilies are useful diagnostically and also have therapeutic potential.

### 3.1. Metalloproteinases

Metalloproteinases (MMPs) are responsible for the irreversible proteolytic destruction of cartilage, especially via the breakdown of type II collagen. Seven matrix metalloproteinases are expressed under varying circumstances in articular cartilage [59,60,61]. Among them, only *MMP*-*1*, *MMP*-*2, MMP*-*13*, and *MMP*-*14* are constitutively expressed in adult cartilage. Their physiological function is tissue turnover and the level of their expression increases significantly in pathologic states. The presence of *MMP*-*3, MMP*-*8*, and *MMP*-*9* in cartilage appears to be characteristic of pathological circumstances only [59]. Additionally, the soluble collagenases *MMP*-*1*, *MMP*-*8*, and *MMP*-*13* play a key role in cartilage destruction. The collagenolytic activity of other MMPs (such as *MMP*-*2* and *MMP*-*14*) is likely minor. It was experimentally demonstrated that *MMP*-*3, MMP*-*9*, and *MMP*-*10* degrade other ECM components, but in vivo, they are unable to cleave native type II collagen [59,62,63]. The proper regulation of expression of the metalloproteinase family depends on many factors and triggers several intracellular signaling pathways. The expression patterns of MMPs in cartilage depend on proinflammatory and pleiotropic cytokines and growth factors [64,65]. The overexpression of MMPs is an important marker of the progression of osteochondral diseases, regardless of etiology [59]. There is a relationship between the increase in MMP expression and the rapid rate of joint destruction [66]. 

MMPs are overexpressed in diseased joints, leading to the disintegration of the ECM, thereby reducing flexibility and resistance to tissue injury [67]. The endogenous inhibitors of MMPs are tissue inhibitors of metalloproteinases (TIMPs); however, in OA activity, they are not effective [61,68,69]. The environment of proinflammatory cytokines such as *IL-1*, *IL-6* and *TNF-α* promotes the increase in MMP expression. Therefore, new therapeutic protocols aimed at restoring joint function are often directed toward the use of small molecule inhibitors of MMP subclasses or inhibitors of the interaction between *IL-1* and its receptor [67,70].

Metalloproteinases are a diverse family of genes for which the correlation of cartilage damage with both the presence of polymorphisms [71] (e.g., –1612 5A/6A polymorphism genotypes of *MMP-3* gene promoter [72] or rs639752, rs520540, and rs602128 in *MMP-3* [73], rs4747096 in *ADAMTS14* [74]), changes in expression patterns (expression changes affecting the development and progression of cartilage degradation are related to mRNA of each *MMP* and *TIMP* [61,75]), and expression modulation through miRNA (e.g., miR-27b and miR-127-5p regulates *MMP-13* expression [76,77], miR-140 regulates *ADAMTS-5* expression [78,79]) has been confirmed. 

Van Meurs et al. proved that active MMPs play an important role in cartilage degradation in both the immune-mediated complex arthritis model and in antigen-induced arthritis models [80,81]. Leong et al. [82] experimentally demonstrated that *MMP-3* expression is dependent on joint mobility. Immobilization of the limb led to an increase in *MMP-3* levels (especially in the superficial zone of articular cartilage), but an hour of daily moderate mechanical stress (passive movement) resulted in a decrease in the growth of *MMP-3* and *ADAMTS-5*. Researchers showed that passive motion loading suppresses immobilization-induced overexpression of the *MMP-3* and *ADAMTS-5* genes, as well as blocking the increase in total articular cartilage *MMP-3* activity. Leong et al. [82] also confirmed that intra-articular injections of the *MMP-3* inhibitor silenced the effects of immobilization and *MMP-3* overexpression. There is no consensus on the scope of the best therapeutic strategy based on targeted enzymes from the metalloproteinases family. Elliot and Cawston [83] point out that it is important to identify the therapeutic benefits of inhibiting proteoglycan or collagen degradation. The most preferred option seems to be the use of a broad spectrum MMP inhibitor that simultaneously inhibits *ADAM* proteinases (e.g., *TACE*, *ADAM17*) [83,84]. 

The level of MMP expression modulation is probably the most important; it is primarily regulated by various growth factors and cytokines [85]. In the context of the results of the research, this family of genes (especially *MMP-3 and MMP-13*) can be considered as the main biomarkers of bone and cartilage damage as well as mediators of joint destruction [23,24,59]. The correlation of *MMP*-*13* expression with cartilage damage makes this gene an interesting candidate for pharmacological intervention [24,86]. 

### 3.2. Growth Factors

Changes in the expression levels of this group of genes are particularly important, not only in the search of biomarkers to monitor the progression of cartilage disease, but also for regenerative therapy. Growth factors are biologically active polypeptides that are used as stimulators of cell growth and enhancers of chondrogenesis, but they can also support the treatment of cartilage defects [87,88]. The most important in terms of supporting therapy include the transforming growth factor (*TGF*) superfamily (e.g., *TGF*-*β1*, *BMP*-*2*, *BMP*-*7*, *TGF*-*β3*, *CDMP*-*1,* and *CDMP*-*2*), fibroblast growth factor (*FGF*) family (e.g., *bFGF*, *FGF*-*2,* and *FGF*-*18*), insulin-like growth factor (*IGF*), and platelet derived growth factor (*PDGF*) [88,89,90,91]. The role of these factors in restoring cartilage tissue function and structure is varied. Growth factors by stimulating anabolic and catabolic processes affect the local microenvironment of the joint. 

Their therapeutic potential in the treatment of cartilage injuries and early arthritis has been demonstrated in both in vitro and animal model studies [87,88,91,92]. *TGF-β* is responsible for stimulation of ECM synthesis, chondrogenesis in the synovial lining, and the reduction of *IL-*1 catabolic activity. *BMP-7* inhibits catabolic factors (e.g., *MMP-1*, *MMP-13*, *IL-1*, *IL-6*, and *IL-8*) and promotes the synthesis of cartilage matrix [87]. Under physiological conditions, *FGF-2* plays a chondroprotective role and is responsible for inhibiting interleukin-1-driven aggrecanase activity [93]. Chia et al. [89] in a mouse model showed that the subcutaneous administration of *FGF-2* inhibited osteoarthritis. At the same time, an increase in OA symptoms was observed in knockout mice of this gene. *FGF-2* suppresses *IL-1-* or *TNF*-stimulated aggrecanase activity in a dose-dependent manner. High doses of *FGF-2* may promote the intensification of inflammation by antagonizing *IGF-1* and increasing *MMPs* [87,93]. *IGF-1* also helps to maintain cartilage integrity, which suppresses anabolic processes that promote cartilage repair [94,95]. The regenerative effect requires cooperation with *TGF-β* and *BMP-7*. As shown in a mouse model, low *IGF-1* expression results in joint cartilage damage, while high *IGF-1* expression promotes synovial membrane protection [90]. The therapeutic application is described in detail later in this review. 

### 3.3. Inflammatory Factors 

The role of synovium and adipose tissue hormones in the intensification of inflammation is emphasized in the development and progression of cartilage diseases. The presence of numerous inflammatory mediators, such as chemokines, interleukins, metalloproteinases, and others, shows a positive correlation with the severity of pain and joint dysfunction. It is now known that the role of inflammation of the tissues forming and surrounding the joint is important for development and progression, and the level of inflammatory response shows a positive correlation with the severity of patients’ complaints [96,97,98]. This creates a potential option for new treatments targeted at modifying transmission pathways dependent on numerous inflammatory mediators.

Based on biochemical and genetic findings [25,26,27], markers of the inflammatory process have been identified. The strongest correlations with inflammation of the joints have been found for (1) inflammatory mediators (cyclooxygenase (*COX*), prostaglandin E2 (*PGE2*), prostaglandin D2 (*PGD2)*, prostaglandin F2a2 (*PGF2a)*, thromboxane and prostaglandin I2 (*PGI2*); (2) circulating or locally occurring cytokines: interleukin-1 (*IL*-*1*), *IL*-*6, IL-17*, *IL*-18, *TNF*-α chemokines, such as C-C motif chemokine ligand 5 (*CCL5*) and *IL-8*; (3) nitric oxide (NO); and (4) synovial degradation products: hyaluronan or hyaluronic acid (HA) [99,100]. Synovial inflammation is also correlated with the secretion of proinflammatory cytokines, including vascular endothelial growth factor (*VEGF*), blood vessel formation (factor VIII), intercellular adhesion molecule-1, and proinflammatory cytokines (*TNF*-α, *IL-6*, and *IL-1ß*) [101]. The consequences of changes within this group of genes and their protein products are primarily increased cartilage degradation and bone resorption, inhibited glycoprotein and collagen synthesis, *MMP* overexpression, the stimulation of other cells to produce proinflammatory cytokines and growth factors, the stimulation of nitric oxide production, and the induction of chondrocyte apoptosis [102]. Therefore, the treatment of this disease should be aimed at reducing inflammation.

Mediators and effectors of active inflammation, most commonly *IL-1ß, IL-6,* and *TNF-α*, are detected in the plasma and synovial fluid of patients with the degenerative disease [96,103,104]. They are released by damaged cells of cartilage, bone tissue, and synovium, which secondarily stimulate the production of other mediators with proinflammatory and catabolic effects on joint structures [105]. Increased chondrocyte apoptosis has been observed in cartilage collected from OA patients, which may be related to their shortened survival [106]. An increase in the concentration of proinflammatory prostaglandins, leukotrienes, interleukins, and metalloproteinases is also observed [107,108]. The presence of inflammation has been shown to inhibit the expression of genes involved in the phenotypic differentiation of chondrocytes, which negatively affects their metabolism and regeneration; therefore, the balance between degenerative and regenerative processes in cartilage tissue is disturbed [109,110]. 

Inflammation is one of the important factors in the progression of cartilage erosion and the intensification of disease symptoms in patients with confirmed OA [111,112], and it is manifested as synovial membrane mononuclear cell infiltration, which can be seen both at the early and late stages [113]. Nevertheless, there is no definitive evidence as to whether inflammation occurs earlier or is a consequence of OA development [101]. At the molecular level, a synovitis marker in a genome-wide study was analyzed by Scanzello et al. The study included material obtained from patients with no symptoms or OA features in the radiological image, who underwent arthroscopic meniscectomy due to knee injuries. The control group consisted of patients with early or advanced OA diagnosis. Scanzello et al. established chemokine expression patterns correlated with the presence or absence of inflammation. The molecular signature of the compared samples differed in the expression of 28 mediators (including *IL-8, CCR7, CCL19, CCL21,* and *CCL5*), most of which were overexpressed in inflammation [101]. 

## 4. Types of Genetic Changes

Characterization of the complex conditions of cartilage damage and the resulting identification of markers useful diagnostically and therapeutically requires the integration of data from genomic, transcriptomic, epigenetic, and proteomic experiments. Single nucleotide polymorphisms (SNPs) that are easy to identify are most useful as diagnostic and prognostic markers. Multiple SNPs in genes associated with cartilage diseases have been identified as risk factors (e.g., OA predisposition, Table 2). The second group of markers comprises genes whose changes in expression level are correlated with the induction and severity of cartilage degradation (described above). These biomarkers and their regulators are primarily useful in developing new therapies and as supplementing existing treatments. Among the regulators of gene expression, microRNAs are of particular interest due to their great therapeutic potential. 

### 4.1. Single Nucleotide Polymorphisms

The effective study of genetic conditions for the initiation and progression of diseases requires progress in understanding their genomic organization; therefore, one of the main directions of research is also the identification of single nucleotide polymorphisms in the structure of selected candidate genes. An interesting research direction is searching for SNPs within the candidate. SNPs, as the most common genetic varieties in the population, have been successfully correlated with the pathogenesis of many diseases of articular cartilage, mainly osteoarthritis (Table 2). 

Multiple SNPs in various genes play different roles in the pathogenesis of OA and its subtypes [123,141,151,152,153,154]. For example, Wang et al. [154] in their overview and meta-analysis described 56 SNPs from different genes that are associated with either hip OA (e.g., in *COL11A1*, *TGF-1β, VEGF*), knee OA (e.g., *IL-6*, *ASPN*, *GDF5*), or both joints (e.g., *IL-8*, *CALM1, SMAD3*) [154]. Guo et al. [73] have experimentally demonstrated that SNPs in the *MMP-3* gene contribute to an increase in OA risk and therefore can serve as molecular markers of osteoarthritis susceptibility. Similar observations were made by Tong et al. [114], who confirmed the relationship of two common allelic variants of *MMP-3* (rs650108) and *TIMP-3* (rs715572) with the risk of developing cartilage disease. Miyamoto et al. [155], Chapman et al. [156], Williams et al. [157], Wang et al. [158], and Southam et al. [159] independently identified a missense mutation (rs143383) in the GDF5 promoter. *GDF5* is regarded as a prominent genetic risk factor associated with OA, and rs143383 SNP significantly reduced the transcriptional activity of *GDF5* in chondrocytes. This change results in the disruption of cartilage synthesis and maintenance by resident cells [159]. The regulation of *GDF5* activity is of therapeutic significance because in scientific and preclinical studies, the effect of this gene on the healing of connective tissue (fibrous and cartilage) has been demonstrated [160,161,162,163]. Parrish et al. [164] showed that intra-articular supplementation of recombinant human *GDF5* (*rhGDF5*) in the osteoarthritic joint leads to a significant slowdown in disease progression.

### 4.2. Epigenetic Changes

The phenotype of mature chondrocytes is stabilized by numerous epigenetic modifications, including DNA methylation (hypo- and hypermethylation mainly in the promoter sequences of target genes at CpG sites), histone modification (methylation, ubiquitination, acetylation, and phosphorylation), and non-coding RNAs binding to the 3′-untranslated region of the target gene [165,166,167,168,169,170,171]. Epigenetic changes have been described for many groups (transcription factors, proteinases, cytokines, chemokines, growth factors, and ECM proteins) of genes relevant for cartilage destruction, hypertrophic chondrocyte formation, and synovitis. Modifications to the epigenetic pattern can lead to genetic disruptions that result in the overexpression of cartilage-degrading proteases and inflammatory process factors [169]. 

In recent years, especially microRNAs have enjoyed increasing interest in the scientific world. These low molecular weight non-coding RNAs are used as diagnostic markers and predictors. However, as post-transcriptional regulators of gene expression, they are an important candidate for personalized therapy [172,173]. Identifying the expression signatures of microRNA sets associated with specific degradation phenotypes opens up new opportunities, both in developing new strategies for preventing tissue structure destruction as well as new options for anti-degenerative therapy [174,175]. MicroRNAs in biofluids are primarily minimally invasive biomarkers that are useful diagnostically and prognostically. Therapeutic strategies based on modulating miRNA activity are increasingly emerging because of the ability of these ncRNAs to affect cell phenotype [173,174,176]. The modulation of expression using microRNA has become a new trend in determining the mechanisms regulating various diseases [177], especially those involving inflammation. Various delivery mechanisms and nano constructions are tested in vivo [79,178,179,180,181], but their routine use in clinical practice should still be seen as a promising future perspective.

Epigenetic regulations of cytokine expression are well described in the literature. For example, *IL-1β* expression levels are modulated by methylation and demethylation of the CpG site of the promoter region [182], as well as the activities of miR-146a and miR-149 [171,183]. In addition, it was found that the use of histone deacetylase inhibitors in chondrocytes results in the reduced secretion of inflammatory cytokines, such as *IL-1β*, suppressing synovitis and preventing the redifferentiation of dedifferentiated chondrocytes [169,184]. An example of research into the therapeutic role of miRNA is the study by Si et al. [79], which explored the possibility of slowing the progression of induced surgically OA in model rats using intra-articular injections of a miRNA-140 mimic or inhibitor. MiRNA-140 is specifically expressed in cartilage, and its role includes the regulation of ECM-degrading enzymes. The overexpression of miRNA-140 in primary human chondrocytes has been shown to promote collagen II expression and inhibit the expression of target genes: *MMP-13* and *ADAMTS-5*. Si et al. [79] observed that in the group treated with miRNA-140 agomir, significantly higher results were obtained in cartilage thickness, chondrocyte count, and collagen II expression level, while simultaneously reducing expression levels of *MMP-13* and *ADAMTS-5*. Thus, the potential of miRNA-140 in the attenuation of OA progression by modulating ECM homeostasis was demonstrated. Similar observations were made by Wang et al. [179] for miR-483-5p (*MMP-2* expression regulator) in a transgenic (TG483) mice model. Researchers, using the appropriate constructs introduced intra-articularly, could both enhance degenerative disease (by introducing LV3-miR-483-5p lentivirus) and delay the progression of experimental OA (using antago-miR-483-5p) [179]. Interesting results have been described by Huang et al. [178], who conducted an intra-articular injection of recombinant AAV for the in vivo overexpression of miR-204 in a surgically induced OA mice model. They showed that two homologous microRNAs, miR-204 and miR-211, are able to maintain joint homeostasis and thus protect the pathogenesis of osteoarthritis. Loss of miR-204 and miR-211 function leads to the accumulation of *Runx2* and induction of proteases that degrade the matrix in articular chondrocytes and synoviocytes. This mechanism activates the destruction of articular cartilage. They showed that intra-articular injection of the miR-204-expressing adeno-associated virus significantly slows the progression of OA.

## 5. Gene Therapy Approaches

Bioactive proteins are difficult to administer effectively; however, gene transfer approaches are being developed to provide their sustained synthesis at sites of repair. The treatment of cartilage lesions is applied by transferring genes, encoding for specific growth factors, into chondrocytes or progenitor cells [185]. Gene delivery into the osteochondral unit is categorized as either in vivo (direct gene delivery into host tissue within the lesion site) or ex vivo (indirect gene delivery, e.g., via stem cells or fibroblasts, following in vitro transfection or transduction). Viral gene vectors (e.g., adenovirus, retrovirus, adeno-associated virus (AAV), and herpes simplex virus) present an efficient method for gene transfer [186]. The direct method is less technically demanding, but indirect gene delivery is safer because the gene manipulation takes place under controlled conditions outside the body. Moreover, safety tests can be performed in the genetically engineered cells before the implantation in the defects. Although the development of effective treatments for osteochondral defects remains complicated, gene transfer might improve repair and regeneration at sites of injury by enabling the local, sustained and, potentially regulated expression of morphogens, growth factors, and anti-inflammatory proteins [187]. Understanding the molecular basis of cartilage and joint diseases is important and useful for the establishment of effective therapies. 

### 5.1. Marrow Stimulation

There are two gene-based techniques of marrow stimulation. Pascher et al. presented an approach to enhance natural repair mechanisms within cartilage lesions by targeting bone marrow-derived cells for genetic modification [188]. As an alternative medium for gene delivery, they investigated the feasibility of using coagulated bone marrow aspirates. Mixing an adenoviral suspension with the fluid phase of freshly aspirated bone marrow resulted in uniform vector dispersion throughout, and rates of transgenic expression in direct proportion to the density of nucleated cells in the corresponding clot. Sieker et al. presented good results with this method using cDNA that encoded bone morphogenetic protein *BMP-2* and Indian hedgehog protein and was effective in improving cartilage repair in osteochondral defects in the trochlea of rabbit knees. However, *BMP-2* treatment carried the risk of intralesional bone formation [189]. Ivkovic et al. used ovine autologous bone marrow transduced with adenoviral vectors containing cDNA for green fluorescent protein or transforming growth factor (*TGF-β1*). The marrow was allowed to clot, forming a gene plug that was implanted into partial-thickness defects created on the medial condyle in the sheep model [190]. This method improved the outcome, and *TGF*-treated defects showed significantly higher amounts of collagen II. In the second approach, the recombinant adeno-associated virus is used directly on the exudate that enters the osteochondral lesion. In a rabbit osteochondral defect model, fibroblast growth factor 2 (*FGF-2*), insulin-like growth factor 1 (*IGF-1*), and the transcription factor *Sox9* have been delivered by the transgene, with promising results [187,191,192,193]. 

### 5.2. Autologous Chondrocyte Implantation

Over the years, Autologous Chondrocyte Implantation (ACI) has been established as a good treatment option to deal with large full-thickness chondral lesions [194,195,196]. This approach requires two surgeries. Firstly, articular cartilage is harvested from a lesser-weight-bearing part of the joint. Then, autologous chondrocytes need to be expanded in culture and implanted into the defect. As the application of ACI has been limited by the high cost of autologous therapy and by the need for two surgeries, using genetically modified allografted chondrocytes could reduce complexity and improve cost-effectiveness. Kang et al. showed for the first time that genetically modified allografted chondrocytes could persist and express transgenes in rabbits’ osteochondral defects [197]. There is also a large body of evidence that confirms that genetically modified allogeneic or autogenous chondrocytes are effective for cartilage repair in animals [90,198]. Ortved et al. presented that transferring *IGF-1* by AAV to autologous chondrocytes improved the repair of full-thickness chondral defects in equines [199]. Genetically enhanced allograft chondrocytes were used in human clinical trials [200]. The transduced cells were surgically introduced into cartilage lesions using a fibrin scaffold. A line of human chondrocytes was obtained from a newborn with polydactyly. One cohort of cells was transduced with a retrovirus carrying *TGF-β1* cDNA. There were no treatment-related serious adverse events. Although knee evaluation scores seemed to indicate a trend toward efficacy, patient numbers were not sufficient to determine statistical significance.

### 5.3. Muscle and Fat Grafts

Evans et al. demonstrated another therapy based on the remarkable potential of genetically modified, autologous skeletal muscle and fat grafts to heal large osseous and chondral defects. These tissues can be harvested, genetically modified, and then press-fit into osteochondral lesions within the time frame of a single surgery. The theory behind muscle-based tissue engineering is related to the unique biology of skeletal muscle-derived cells. Skeletal muscle contains satellite cells that are capable of fusing to form post-mitotic, multinucleated myotubes and myofibers. The post-mitotic myofibers are stable cells and theoretically capable of long-lasting gene expression. Their potency is likely to reflect the presence of endogenous progenitor cells, the secretion of morphogenetic signals by the genetically modified cells, and the scaffolding properties of the tissues themselves. When compared to ACI, the complexity of the procedure is reduced, which should lower costs. Moreover, skeletal muscle and fat are easily accessible and available for eventual biopsies. Results from pilot experiments with rabbits showed that using adenovirus vectors carrying *BMP2* cDNA is encouraging. The implanted genetically modified fat and muscle grafts formed bone in the subchondral region and cartilage above, indicating the impact of the progenitor cell location on the process of differentiation [201].

### 5.4. Scaffolds

The main problem to the treatment of focal cartilage defects with the non-scaffolds approach is that the genetically modified cells or gene vectors are diluted by the joint fluid and fail to maintain at the target lesion area [202]. To avoid this obstacle, a promising approach is to deliver modified cells or gene vectors using different types of biomaterials. Scaffolds for cartilage repair present new options to structurally support cartilage repair [203]. Gene therapy, combined with scaffolds, increases the efficiency and duration of transfected genes, forming an efficient system to promote osteochondral regeneration. When the scaffold is degraded, the contents are slowly released to the target area. These biomaterials can be implanted into articular cartilage defects to provide gene transfer that enables the controlled release of the vector over time. The regulated transmission of genetic material via biomaterials could enhance the properties of the gene products and protect these active agents against degradation [204]. Additionally, biomaterial-mediated gene delivery provides biomechanical properties in magnitudes similar to that of native articular cartilage, which is especially important for the repair of large defects [205]. Using scaffolds in gene therapy for osteochondral tissue repair can be applied by two methods: through the incorporation of the vector during scaffold preparation and putting into the chondral lesion to transfect chondrocytes (Figure 2) or by the connection of the vector with cells to a formed scaffold. 

Among the numerous synthetic materials, biocompatible and biodegradable compounds matrices are thought to possess the most promising potential for repair of osteochondral defects [206,207,208]. Different gene delivery approaches include the use of solid scaffolds, hydrogels and micelles (alginate, poloxamer PF68, and poloxamine T908 polymeric micelles based on poly(ethylene oxide)—PEO—and poly(propylene oxide)—PPO—triblock copolymers, self-assembling RAD16-I peptide hydrogels, polypseudorotaxane gels) to create vector-loaded biomaterials [186,209,210]. These biomaterials provide the formation of cartilage tissue with adapted mechanical properties and affording protection against tissue degradation in conditions that enable joint resurfacing. Different scaffold-based approaches are shown in Table 3, taking into account the vector, biomaterial, and transferred genes.

### 5.5. Mesenchymal Stem Cells

The use of controlled gene delivery approaches to facilitate clinical cartilage repair is still a developing area. Emerging approaches include the use of progenitor cells, rather than chondrocytes, as agents of gene transfer for cartilage repair. Mesenchymal stem cells (MSCs) are the most widely studied due to their high availability and proliferative/differentiation ability. Bone marrow-derived MSCs (BMSCs) and adipose-derived MSCs (ASCs) are commonly employed for osteochondral therapy [210]. Although genetically modified MSCs are not being used in human clinical practice, some experimental studies have been completed. Leng et al. used transfected BMSCs with hIGF-1 cDNA and mixed with calcium alginate gels for transplantation into osteochondral defects on the femoral condyle of goats and showed the improvement in the repair [220]. Yang and al. transfected BMCs with adenoviruses expressing C-type natriuretic peptides and seeded the cells onto silk/chitosan scaffolds to promote chondrogenesis in rats [221].

In another study, Venkatesan et al. designed 3D fibrin–polyurethane scaffolds in a hydrodynamic environment that provided a favorable growth environment for recombinant adeno-associated virus vector (rAAV)-infected *Sox9*-modified human BMSCs and promoted their differentiation into chondrocytes [222]. 

Controlled tissue growth and biomimetic cartilage properties were maintained upon seeding the human ASCs into large PCL scaffolds immobilized with Dox-inducible lentiviruses expressing *IL-1Ra* [205]. Although these techniques have not been confirmed in clinical studies yet, they hold great scientific promise for treating cartilage injuries in the near future. In addition, interest in improving the efficiency and targeting of non-viral vectors continues. For example, Pi et al. identified a chondrocyte-affinity peptide that enhances cartilage-targeting transfection when attached to polyethyleneimine in rabbit knees [223]. These data demonstrate that the potential for gene therapy of cartilage lesions is encouraging; however, the area is still under development. The assessments of possible associated toxicities are also lacking, but they are essential for clinical translation.

## 6. Conclusions and Future Directions

Genes and their polymorphisms play a key role in the development of the osteochondral pathologies in the general population. The complex background of the genetic heterogeneity of the cartilage diseases consists of a plethora of genes and their epigenetic modifications. It results in changes in their activity, and thus explains the backgrounds of pathological processes. Analyses of allelic expression imbalances, genetic expression signatures, epigenetic regulatory mechanisms, and key cellular pathway disturbances provide comprehensive knowledge that brings us closer to understanding normal tissue metabolism as well as pathological processes. Therefore, the challenge is to find sensitive and specific biomarkers that will help in the early diagnosis and targeted therapy of cartilage ailments. Genetic engineering, a combination of gene transfer techniques and tissue engineering, is one of the potential new strategies for the treatment of osteochondral injuries. The potential advantage of gene transfer into the chondrogenic cells relies on sustained levels of growth factors, which can be reached through transgene expression in situ. Although the field of genetic engineering is young, current research offers the gene transfer approaches developed to provide a sustained synthesis of bioactive reagents at the cartilage repair sites. To augment the regeneration of articular cartilage, therapeutic genes can be delivered directly to the cartilage lesion. Since cartilage injuries are not life-threatening, the safety of gene transfer approaches for repair is of particular importance. Therefore, the harnessing of this technology for clinical use is strongly dependent on the use of safe and efficient vectors, transgenes, and delivery systems. The major considerations for the clinical translation of such methods are their biology, safety, ease of manufacture, and cost-effectiveness [224]. In this regard, combining gene therapy with tissue engineering concepts might overcome the various physiological barriers that impede the safe, effective, and long-term treatment of damaged articular cartilage. Cartilage repair could become a significant domain of gene therapy due to the pervasiveness of osteochondral pathologies, and applications in gene therapy may provide local treatment with a relatively small amount of vector.

## Figures and Tables

**Figure 1 ijms-21-05430-f001:**
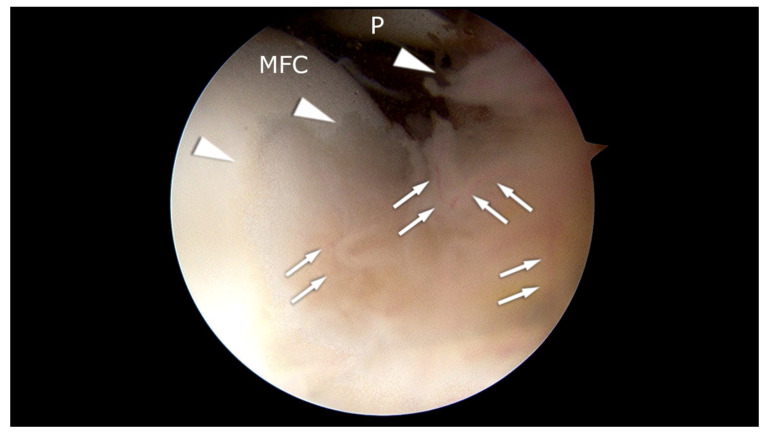
Arthroscopic appearance of the patient with synovitis and initial pathologic changes in the cartilage of the medial femoral condyle (MFC). Arrowheads: hypertrophic synovium. MFC: cartilage of the medial femoral condyle. P: patella. Arrows: blood vessels. The picture comes from our own material.

**Figure 2 ijms-21-05430-f002:**
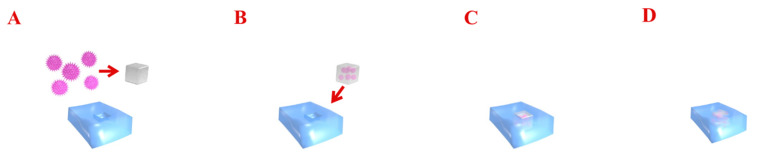
The diagram presents gene delivery therapy into cartilage. Transferred genes encoding the specific growth factors are combined with viral gene vectors (**A**,**B**). Such a construct is implanted into the chondral defect (**C**). Gene vectors are able to transfect chondrocytes and improve regeneration at sites of injury by enabling the local expression of morphogens, growth factors, and anti-inflammatory proteins (**D**).

**Table 1 ijms-21-05430-t001:** Summary of biomarkers of cartilage damage showing the potential to support regenerative therapy. ECM: extracellular matrix.

Gene Category	Gene(s)	Therapeutic Potential	Ref.
extracellular matrix components	*PIIANP, CTX-II, PIINP, COMP, type II collagen*	inhibition of collagen and proteoglycan degradation, stimulation of cartilage matrix synthesis and/or cell proliferation	[13]
matrix-degrading enzymes proteolytic enzymes	metalloproteinases (MMPs such as *MMP*-*3*, *MMP*-*9* and *MMP*-*13*); aggrecanases (e.g., *ADAMTS*-4 or *ADAMTS*-5) and their inhibitors (*TIMP*-*1*, *TIMP* -*2*)	inhibition of ECM degradation	[14,15,16,17,18,19,20,21,22,23,24]
circulating or locally occurring cytokines, inflammatory mediators and anabolic growth factors	*TGF-β1,2,3*; *BMP-2,-4,-7*; *CDMP-1,-2,-3*; *IGF*-*1*, *PDGF*, *EGF*, *HGF*	stimulation of chondrogenic differentiation, cartilage matrix synthesis, and/or cell proliferation	[25,26,27]
inflammatory factors	*IL-1Ra, IL-1R, ICE inhibitor, TNF-R, anti-TNF-antibodies, TACE inhibitor, TIMP-1,-2, MMP inhibitors, IL-4,-10,-11,-13*	anti-inflammatory mechanism: *IL-1* blockage, *TNF-α* and MMPs inhibition	[21,28,29,30,31,32,33,34,35]
transcription factors	*Runx2*, *CEBPβ*, *HIF2α*, *Sox4*, *Sox9,* and *Sox11*	stimulation of chondrogenic differentiation	[36,37,38,39,40]
growth factors and signal transduction molecules	*PTHrP, IHH, SHH, DHH, Smad 6, -7, mLAP-1*	inhibition of osteogenesis/hypertrophy, *TGF-β/BMP* action, and terminal differentiation	[41,42,43,44,45,46]
apoptosis regulators	*Bcl-2, Bcl-XL, Anti-FasL, Akt, PI3-kinase, NF-κB*	apoptosis inhibition: caspase inhibition, FAS-l blockage no-induced apoptosis, *TNF-α* and trail inhibition	[47,48,49,50]

**Table 2 ijms-21-05430-t002:** Polymorphism from selected genes correlated with the risk of osteoarthritis.

Gene	SNPs	Affected Joint and Impact	Reference
***MMP-3***	rs639752, rs520540, rs602128, rs650108, rs679620	increased risk of hip, knee, hand OA breakdown of extra-cellular matrix proteins	[73,114]
***MMP-8***	rs1940475, rs3765620	common allelic variants associate with knee and hand OA predisposition (conflicting or inconclusive)	[115]
***TIMP-3***	rs715572 (G/A), rs1962223 (G/C)	association with hip, knee OA risk in the recessive model rs715572G/A could be a candidate protective gene for severe knee OA interactions between *SMAD3* rs6494629 T/C and *TIMP3* rs715572 G/A polymorphisms coul have protective roles in knee OA	[114,116]
***COL11A1***	rs1241164, rs4907986, rs2615977	association with hip OA risk	[117]
***COL9A2***	rs7533552	asociation with hip OA G allele in COL9A2 changes a glutamine to arginine or to tryptophan (changes a polar and thus important change in load-bearing cartilages stabilizing the collagen fibrils)	[118]
***VEGF***	rs833058	association with hip OA in men	[117]
***DVWA***	rs7639618, rs9864422, rs11718863	risk factor for developmental dysplasia of the hip and knee etiology (conflicting or inconclusive)	[119,120,121]
***FRZB***	rs7775, rs288326	risk factor for hip, knee and hand OA (conflicting or inconclusive) relationship between proximal femur shape and incident radiographic hip OA (rs288326)	[122,123]
***GDF5***	rs143383 (risk allele T)	risk factor for hip, knee and hand OA (conflicting or inconclusive)	[123,124]
***CALM1***	rs12885713 *(-16C/T transition SNP)*	association with the pathogenesis of hip and knee OA (conflicting or inconclusive, probably dependent on populations) −16T allele decreases CALM1 transcription in vitro and in vivo correlation with articular chondrogenesis regulation and adhesion of chondrocytes to extracellular matrix proteins during the cartilage repairing process	[125,126,127,128]
***TGF-1β***	rs1982073, rs1800470 (TT genotype), rs180046 (TT genotype and T allele), rs1800469 (TT genotype and T allele)	increase the individual susceptible of OA contribution in disturbances of homeostasis and cartilage formation	[116,129]
***IGF1***	rs2195239, rs11247361 (4488C>G)	increased risk of knee, hip, hand, and spine OA osteoarthritic cartilage degradation	[130,131,132,133]
***IL1RN***	rs9005, rs315952, rs419598, rs315943, rs315920 (C/t)	risk factor for hip, knee, and hand OA (conflicting or inconclusive) marker of progression of knee and hip OA	[134,135,136,137,138,139]
***IL-6***	rs1800795 (−174G>C)	association with hip OA (conflicting or inconclusive)	[132,140]
***MCF2L***	rs11842874 (risk allele G)	risk factor for knee OA association with reduction in pain and improvement in function for knee OA patients	[141,142,143]
***SMAD3***	rs12901499 (GA and GG genotypes), rs12102171 (CC and CT+TT and TT and TG+GG), rs2289263, rs6494629 (T/C),	together with body mass index (BMI) may be susceptible risk factors to OA interactions between *SMAD3* rs6494629 T/C and *TIMP3* rs715572 G/A polymorphisms could have protective roles in knee OA	[116,144,145]
***BMP5***	rs921126 (GA and GG genotypes)	significantly increased risk of knee OA	[145]
***ADAM12***	rs1871054, rs3740199 (Gly48Arg)	association with female knee OA (conflicting or inconclusive) candidate for the prevalence and progression of knee OA	[132,146]
***ADAMTS14***	rs4747096	association with knee OA and Achilles tendon pathology	[147,148,149,150]

**Table 3 ijms-21-05430-t003:** Scaffold-based gene therapy approaches in cartilage repair.

Gene(s)	Vector(s)	Scaffold	Reference
*Sox9*	rAAV	polymeric micelles	[211]
	rAAV	poloxamers, poloxamines, micellar systems	[211]
*IL-1Ra*	lentiviral	poly-e-caprolactone	[202,205,209]
*lacZ*	rAAV	poloxamers, poloxamines, micellar systems	[207]
	rAAV	alginate, alginate/poloxamers	[212]
	rAAV	polypseudorotaxane gels	[213]
*TGF-β1*	rAAV	poloxamers, poloxamines, micellar systems	[214]
	lentiviral	poly-e-caprolactone	[215]
	rAAV	polymeric micelles	[214]
*IL1RN and IGF1*	non-viral	chitosan	[216]
*lacZ, RFP*	rAAV	self-assembling peptide hydrogels	[212,217]
*lacZ, RFP, Sox9*	rAAV	poloxamers, poloxamines, micellar systems	[213,218]
*Sox9, TGF-β*	rAAV	carbon dots	[219]

rAAV: recombinant adeno-associated virus vector; lacZ: E. coli β-galactosidase; RFP: red fluorescent protein; Sox9: sex-determining region Y-box 9; TGF-β: transforming growth factor beta; IL-1Ra: interleukin 1 receptor antagonist.

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
