# Peer review of "Genetics in Cartilage Lesions: Basic Science and Therapy Approaches"

_ijms, 2020, doi:10.3390/ijms21155430_

Round 1
Reviewer 1 Report
IJMS-832403 Genetics in cartilage lesions: from basic science to clinical practice
Overview: This is a review article that presents some of the known information about gene expression in cartilage and synovium as well as potential opportunities for gene therapy in the treatment of osteochondral disease.
General Comments: The overall organization, as well as the organization of points within individual sections, feels somewhat scattered. In some places, major concepts are glossed over, while in others unnecessary detail is included, muddling the main point. A major concern is that a large number of the citations are review articles rather than original literature. A review should distill the primary literature to elucidate major points, which has not been done consistently in this manuscript. There is odd syntax and word choices throughout, which I suspect may be due to inaccurate translation to English. Some of the most egregious examples are listed in the specific comments below. Review and revision by a native English speaker familiar with scientific writing and this topic would be strongly recommended.
Without line numbers in the manuscript, I have tried to reference my comments by section, page, and paragraph.
Introduction
Page 1, bottom paragraph: “serine and cistern prostheses” Please explain what you mean by this. I suspect this may be a translation error?
Page 2, top paragraph: “reconstruction” This is an odd word choice – I believe you mean “turnover”. The following sentence should begin “Proinflammatory cytokines…”
Genetic Basis of Cartilage Pathology
Page 2, bottom paragraph: “The genetic lesion landscape…” This entire sentence does not make sense as written and needs to be re-worked.
Figure 1: I am unsure why you have chosen to lay out your “schematic” in this way. For example, on the right-hand side the arrows would suggest that you are claiming that in the synovium IL-1 stimulates TNF which stimulates NO. Is this really what you mean? How does this relate to what you are showing in the “cartilage” schematic below? On the left-hand side, are you trying to show that inflammation and MMP/TIMP synthesis occur independent of each other? The figure is not referenced anywhere in the text and in its current form does not really add anything.
Page 3, top half of page: These lists are difficult to read and would probably be better in a table. Also, not all of them have citations.
Page 3, middle paragraph: “In clinical practice…” This sentence needs to be appropriately cited.
Metalloproteinases
Page 3, second to bottom paragraph: “Overexpression of MMPs…” This and the sentences that follow at the end of this paragraph all need to be appropriately cited.
Page 3, bottom paragraph: This entire paragraph is an awkwardly phrased run-on sentence. Interesting information is here, but it is glossed over rather than being described appropriately.
Synovitis
Page 4, first full paragraph: “Their activation occurs due to the action of inflammatory mediators…” You need to clarify this statement (and provide appropriate citation). Do you mean that circulating inflammatory mediators induce synoviocytes to produce their own products, or do you mean that products released by the cartilage result in this? There is evidence that suggests that synovitis is an initiating factor in osteoarthritis – that is, that it preceded cartilage damage.
Figure 2: Need to be consistent with the use of MFC (more standard and therefore preferred) and MCF. Also, it is interesting that you have chosen to show an image of a joint without evidence of synovitis in the section about synovitis. A different image would illustrate your point better. As it is, it is not particularly useful.
Page 4, middle of page: Again, this listing of biomarkers/genes is difficult to read and would probably be better in a table.
Page 4, 3rd paragraph: “Synovial biopsy tests…” You need to be more specific about the sources of these synovial biopsies. Were these from normal joints, early OA, end-stage OA? What do you mean by “differentiated expression” (i.e. what conditions are being compared)? This is extremely important for interpretation of the data you are presenting.
Single Nucleotide Polymorphisms
Page 5, top of page: Again, a table would be more useful, and would allow identification of the appropriate primary source with each SNP reported. More importantly, it is crucial to differentiate between a SNP being associated with a disease (i.e. in a GWAS) and a SNP being implicated in disease pathogenesis. There have been many associations reported, but very little evidence to show direct mechanistic roles of individual variants in disease onset and progression.
Epigenetics and MicroRNAs: These sections seem to have been written by two different people. The epigenetics section is a confused glossing over of the topic, while the microRNA section has specific examples from the primary literature upon which conclusions have been based. The latter approach is by far preferred.
Gene Therapy Approaches
Page 6, top paragraph: “…effective reagents for osteochondral defects…” I’m not sure if you mean “treatments”? “Reagents” does not make any sense.
Page 6, 2nd paragraph: This entire paragraph has nothing to do with gene therapy approaches. Why is it included here?
Marrow stimulation: “The effectiveness of microfracture…” None of the techniques you describe here have anything to do with microfracture and this sentence needs to be removed.
Scaffolds
Page 7, bottom paragraph: “The main problem…” This sentence needs a citation.
Page 7, bottom paragraph: “biomaterial-mediated gene delivery provides…” This sentence needs a citation.
Figure 3: You state in the text that there are two methods for using gene therapy and scaffolds together, but your figure does not indicate which of these you are trying to demonstrate. Further, are you trying to demonstrate that the viruses are directly incorporated into biomaterials without any kind of cell seeding? This is really important since the viral vectors don’t produce any products from the transgene without cellular machinery. Do the viruses exit the scaffold once implanted and transfect the native tissue? This needs to be more clear in the text. I’m not sure the figure adds much.
Author Response
Dear Reviewer,
we are grateful to You for considering this manuscript for publication. Indeed, the original comments and remarks made by You and the and second Reviewer helped us to significantly improve the manuscript. A point-by-point response is appended to this letter and as You suggest all the sentences highlighted in the English language editing report were rephrased.
We very much appreciate Your willingness to review the manuscript in light of the further information now provided and to ascertain if our manuscript is suitable for publication in the International Journal of Molecular Sciences.
I appreciate You again for Your thoughtful criticism and kind advice. Your review greatly helped to refine our manuscript. We accepted most of Your comments and have done our best to improve our manuscript. I look forward to hearing from You soon.
With best regards,
Joanna Szczepanek, Ph.D.
Centre for Modern Interdisciplinary Technologies

Reviewer 2 Report
The review article topic is interesting but the write up is not very well articulated. Lacks strong citations of reference from where the reported data or analysis is conferred. The review requires good structuring with language revisions. Few additional data is also recommended.
- The title is wider but the article does not essentially provide that information. Suggest simplifying the title as per the content of the present review.
- The abstract needs little me addition of what the present review highlights and would like to convey a conclusion based on the literature and clinical analysis.
- Section “GENETIC BASIS OF CARTILAGE PATHOLOGY” requires an extensive reference citation.
- The source from which the schematic fig 1 is prepared should be cited properly.
- Introduction under “synovitis” lacks citation of references.
- Fig 2 lacks reference from where is has been adopted. Permission from publisher to be reproduced in this article.
- The review lacks to interrelate or correlate on how the genes or micro RNA or epigenetics can be translated into therapeutics instead of just as biomarkers for disease identification.
- Schematic figures for gene therapy or marrow stimulation and other approaches are encouraged because it can help the reviews to get a more meaningful understanding.
- A tabulated review having more reports on scaffold-based therapies is recommended. Scaffold based therapies are upcoming and advancing strategies and largely translated into clinical settings by biomedical companies.
- Section MSC- should be rewritten as Mesenchymal stem cells (MSCs).
- A section of outlook and future perspectives of gene therapies is recommended.
- The conclusion should be restructuring mentioning gene therapies require a lot more clinical experimentation those are challenging for generating data required for regulatory bodies for approvals of these strategies.
Author Response

(The authors gave the same response as above.)

Round 2
Reviewer 1 Report
IJMS-832403 (Revision) Genetics in cartilage lesions: from basic science to clinical practice
Overview: This is a review article that presents some of the known information about gene expression in cartilage and synovium as well as potential opportunities for gene therapy in the treatment of osteochondral disease.
General Comments: While tracking comments is helpful for identifying where major changes were made, it makes the document extremely difficult to read. I strongly recommend always including a “clean copy” of the revised version for ease of reading. Having said that, this manuscript is much improved over the previous version, both in terms of organization and quality of English. In the edited portions, some occasional odd word choices and grammar errors remain, but these can be addressed by careful copyediting. The brand new sections are not as well-written – not sure if they were edited by the same person as the rest of the manuscript? The language editing seemed less robust as the manuscript progressed and at least one instance of near-plagiarism was found. There has been an effort made to bring in more primary sources into the manuscript, although the flow from general comments to specific examples is not seamless. In many cases, new information seems to have just been inserted without consideration for logical progression of concepts. Specific comments follow.
Without line numbers in the manuscript, I have tried to reference my comments by page and paragraph.
Introduction
Page 2, 2nd paragraph: “Proinflammatory cytokines…participate in the treatment of the cartilaginous matrix…” I think perhaps you mean “participate in the turnover of the cartilaginous matrix”?
Genetic Basis of Cartilage Pathology
Page 2, last paragraph: The molecular biology techniques you list are correct but outdated. Most modern studies use next-generation sequencing technologies (whole-genome sequencing, RNAseq, exome sequencing, etc.) to identify variants of interest and evaluate gene expression changes. These newer techniques should be added.
Page 2, first paragraph: you use SNP here, but have not yet defined it – fine to say “…structural changes within DNA (mainly single nucleotide polymorphisms [SNPs])…”
Page 4, first paragraph below the table: Avoid generalities whenever more specific information is available. In this paragraph you state, “For several signaling pathways…and transcription factors…important implications in cartilage lesions were determined.” This tells me nothing. What were the implications? Were they associated with a specific disease, with evidence of increased tissue turnover across diseases, something else?
Page 4, first paragraph below the table: A new paragraph should be started with the sentence “Cartilage diseases are often accompanied by synovitis.”
Page 4, first paragraph below the table: You state, “Their activation occurs due to the action of inflammatory mediators and cartilage matrix molecules…” Please be more specific. Are these released from damaged cartilage? Present in the synovial fluid? Systemic? Or some combination of these? This is a very important concept in our current understanding of the pathogenesis of osteoarthritis. The final sentence of this paragraph regarding clinical and radiographic changes should be moved earlier, prior to the sentences about synovitis.
Metalloproteinases
Page 5, bottom paragraph: “It indicates this family of genes…” This does not make sense as written. Do you mean, “It implicates this family of genes…”?
Page 5, bottom paragraph: Please start a new paragraph with the sentence “MMPs are overexpressed in diseased joints…”
Page 6: You have inserted new, more detailed, information in the middle of the MMP section, but this leaves the last paragraph of this section hanging. This section should be reorganized so that the information included in this last paragraph is moved back up to the more general information. As it is, it just feels disconnected.
Growth Factors
Page 6, bottom paragraph: “As researchers point out…” This should be simplified to just state, “FGF-2 suppresses IL-1 or TNF-stimulated aggrecanase activity in a dose-dependent manner.” An appropriate citation should then be added.
Page 7, top paragraph: “As shown in the mouse model…” should either be modified to state which model was used, or simply modified to say “As shown in a mouse model…”
Inflammatory Factors:
Page 7, first paragraph of this section: The shifting tense here is confusing and the paragraph should be reorganized. In one place you state that the role of inflammation is not clear-cut, and several sentences later (indeed, throughout the rest of this section) you state that inflammation is important for disease development and progression. This needs to be tightened up.
Page 7, bottom paragraph: “…markers of the inflammatory process have been selected.” Selected for what? Please clarify. Also, if you are going to leave these lists, either incorporate it into the paragraph as normal sentences, or make it more clear that you are deliberately including a list (e.g. with bullet points).
Page 8, first full paragraph: Not sure what you mean by “Exponents of active inflammation…” Do you mean markers of active inflammation?
Single Nucleotide Polymorphisms
Page 8, just prior to table: The sentence “more than fifty SNPs…” is a nearly exact copy from the Wang et al. article that is cited. This is plagiarism and needs to be rewritten.
Table 2: You cite your main reference for this table as the Wang et al meta-analysis. However, not all of the SNPs listed in this table are found in this reference (for example, those in MMP3 and TIMP3, neither of which are mentioned in the Wang paper). If additional resources were used, then they should be cited appropriately. Further, it is incorrect to state that Wang identified these SNPs – in fact, this group summarized the findings from 7 other studies that were determined to meet the inclusion criteria for their meta-analysis. Additionally, their meta-analysis did not uphold the associations reported in the original studies for nearly all of these SNPs. When reporting results from genetic studies, it is crucial to differentiate between statistical association and mechanistic results. You have not provided that context in this section.
Epigenetic Changes
Page 10: The reorganization of this section is somewhat confusing. You have moved part of a paragraph about miRNA up, but then follow it by a mixed paragraph talking about methylation and miRNA, and then follow that by a long and very detailed section on specific miRNAs again.
Gene Therapy Approaches
Marrow stimulation
Page 11, bottom paragraph: The first sentence (“The effectiveness…”) does not make grammatical sense and I’m not sure what it is referring to. You don’t actually explain what you mean by marrow stimulation – are you talking about micropicking, or something else?
ACI
The end of this section feels incomplete. You state that a cohort of cells was transduced, but don’t state what the outcome was. Are you simply stating that it can be done? Please clarify.
Muscle and Fat Grafts
In contrast to the rest of this section, you do not provide concrete examples of gene therapy for this section. Without this information, it doesn’t really fall under “Gene Therapy Approaches”.
Scaffolds
Figure 3: I cannot see that there is a difference between C and D in the new figure. The description in the text of the figure legend describes the process as well or better than the figure itself. Unless changed significantly, this figure is unnecessary.
Page 13, paragraph above the table: The edited first sentence no longer makes sense as written. The sentence at the end of this paragraph can also be significantly shortened for clarity: “Table 3 presents different scaffold-based approaches that have been reported.”
Author Response
Dear Reviewer
Thank you for Your comments and constructive suggestions, which have been addressed in our point-point response below. The relevant changes have all been incorporated into the revised version as specified in each response, and marked in blue fonts.
We hope you agree that the comments raised by the Reviewer have been appropriately addressed, allowing the revised manuscript to be suitable for publication in the International Journal of Molecular Sciences.
Yours sincerely,
Joanna Szczepanek
On behalf of all co-authors
Overview: This is a review article that presents some of the known information about gene expression in cartilage and synovium as well as potential opportunities for gene therapy in the treatment of osteochondral disease.
General Comments: While tracking comments is helpful for identifying where major changes were made, it makes the document extremely difficult to read. I strongly recommend always including a “clean copy” of the revised version for ease of reading. Having said that, this manuscript is much improved over the previous version, both in terms of organization and quality of English. In the edited portions, some occasional odd word choices and grammar errors remain, but these can be addressed by careful copyediting. The brand new sections are not as well-written – not sure if they were edited by the same person as the rest of the manuscript? The language editing seemed less robust as the manuscript progressed and at least one instance of near-plagiarism was found. There has been an effort made to bring in more primary sources into the manuscript, although the flow from general comments to specific examples is not seamless. In many cases, new information seems to have just been inserted without consideration for logical progression of concepts. Specific comments follow.
Thank you for Your comments. Improving the previous version of the manuscript we suggested above all the comments of the Reviewers. As suggested, we imposed a style that the reviewer suggested as recommended. In this version of the manuscript, we have paid more attention to the seamlessly of the text and its organization again. Comment tracking is the editor's recommendation: „Any revisions should be clearly highlighted, for example using the "Track Changes" function in Microsoft Word, so that changes are easily visible to the editors and reviewers.”This is why we leave this form of editing. In addition, we used the blue font color to highlight the latest changes.
Without line numbers in the manuscript, I have tried to reference my comments by page and paragraph.
Comment: We've numbered the lines for ease in the next review.
Introduction
Page 2, 2nd paragraph: “Proinflammatory cytokines…participate in the treatment of the cartilaginous matrix…” I think perhaps you mean “participate in the turnover of the cartilaginous matrix”?
Comment: The change has been made.
Genetic Basis of Cartilage Pathology
Page 2, last paragraph: The molecular biology techniques you list are correct but outdated. Most modern studies use next-generation sequencing technologies (whole-genome sequencing, RNAseq, exome sequencing, etc.) to identify variants of interest and evaluate gene expression changes. These newer techniques should be added.
Comment: For many years there have been a variety of technologies and tools used in functional genome analysis. In this sentence, we list only examples of techniques most commonly used to identify new biomarkers. I agree that high-throughput methods, especially sequencing, revolutionized the approach to identify and quantify genes and their regulators, and elucidate their function, which is why we added the sentences: “The advent of sequencing methodologies (next-generation sequencing or massively parallel high-throughput DNA sequencing) has provided new possibilities for searching, quantitative and functional analysis of genome and transcriptome. Main purposes of large-scale genomic screens are searching for novel biomarkers and their genetic signatures, and also associated with function in physiological and pathological conditions.”
Page 2, first paragraph: you use SNP here, but have not yet defined it – fine to say “…structural changes within DNA (mainly single nucleotide polymorphisms [SNPs])…”
Comment: The change has been made.
Page 4, first paragraph below the table: Avoid generalities whenever more specific information is available. In this paragraph you state, “For several signaling pathways…and transcription factors…important implications in cartilage lesions were determined.” This tells me nothing. What were the implications? Were they associated with a specific disease, with evidence of increased tissue turnover across diseases, something else?
Comment: In our assumption, this sentence was of an introductory character and in the following we provide details of the impact of disorders on the development of cartilage diseases and therapeutic implications. Nevertheless, according to your suggestion, we have slightly rewritten and specified this fragment of the text. We are aware that the information is presented briefly here, but we do not want to expand the manuscript by the next chapter. We do not exhaust the topic, but the manuscript also has its limitations. We added the sentences: Quantitative and qualitative changes in key genes trigger a cascade of changes that lead to disorders in several signaling pathways (like signal transduction, NF-ĸB, JAK/STAT, Wnt and mTOR)[51]. The initiation and progression of osteoarthritis may be associated with, inter alia, changes in IL-1β expression levels that affect signal transduction through the NF-ĸB pathway [52]. Cartilage degradation by the proteasome-ubiquitin system and intra-cartilage ossification have been correlated with abnormalities in the Wnt pathway mediated by Sox9 and FRZB [53, 54]. In turn, genetic changes in IL-1, IL-6 and TNFα affect, via the NF-ĸB and JAK/STAT pathways, the induction of rheumatoid arthritis [55, 56]. Therefore, the introduction of inhibitors of overexpressed transcription factors and proimflammatory cytokines may have clinical benefits in the regulation of chondrocyte proliferation and differentiation [36].”
Page 4, first paragraph below the table: A new paragraph should be started with the sentence “Cartilage diseases are often accompanied by synovitis.”
Comment: The change has been made.
Page 4, first paragraph below the table: You state, “Their activation occurs due to the action of inflammatory mediators and cartilage matrix molecules…” Please be more specific. Are these released from damaged cartilage? Present in the synovial fluid? Systemic? Or some combination of these? This is a very important concept in our current understanding of the pathogenesis of osteoarthritis. The final sentence of this paragraph regarding clinical and radiographic changes should be moved earlier, prior to the sentences about synovitis.
Comment: The last sentence has been moved to the penultimate paragraph. Other information in this paragraph has an introductory character, and their continuation and details are described in the subsection "Inflammatory factors". The manuscript is currently multi-threaded and due to the size limitations of the text, we are not able to develop every aspect of cartilage diseases. We are aware that the topic is extensive and there are many factors affecting the pathogenesis and progression of diseases, however, our assumption is to present as detailed as possible aspects of genetic changes that have implications primarily in therapy.
Metalloproteinases
Page 5, bottom paragraph: “It indicates this family of genes…” This does not make sense as written. Do you mean, “It implicates this family of genes…”?
Comment: We meant "indicating", but we have rewritten the sentence for better understanding. Now is: “In the context of the results of the research, this family of genes (especially MMP-13) can be considered as the main biomarkers of bone and cartilage damage as well as mediators of joint destruction.”
Page 5, bottom paragraph: Please start a new paragraph with the sentence “MMPs are overexpressed in diseased joints…”
Comment: The change has been made.
Page 6: You have inserted new, more detailed, information in the middle of the MMP section, but this leaves the last paragraph of this section hanging. This section should be reorganized so that the information included in this last paragraph is moved back up to the more general information. As it is, it just feels disconnected.
Comment: We treated the paragraph as a summary, indicating further genetic changes affecting the activity of metalloproteinases in cartilage (which due to limitations cannot be described in detail). Nevertheless, according to Your suggestion, we moved above.
Growth Factors
Page 6, bottom paragraph: “As researchers point out…” This should be simplified to just state, “FGF-2 suppresses IL-1 or TNF-stimulated aggrecanase activity in a dose-dependent manner.” An appropriate citation should then be added.
Comment: The sentence has been shortened as suggested. The citation is already posted after the next sentence, which describes how the activity is related to the dose.. Therefore, in our opinion, there is no need to duplicate the citation.
Page 7, top paragraph: “As shown in the mouse model…” should either be modified to state which model was used, or simply modified to say “As shown in a mouse model…”
Comment: The change has been made.
Inflammatory Factors:
Page 7, first paragraph of this section: The shifting tense here is confusing and the paragraph should be reorganized. In one place you state that the role of inflammation is not clear-cut, and several sentences later (indeed, throughout the rest of this section) you state that inflammation is important for disease development and progression. This needs to be tightened up.
Comment: Thank you for the comment. We removed the first 2 sentences that present a "historical" approach to inflammation.
Page 7, bottom paragraph: “…markers of the inflammatory process have been selected.” Selected for what? Please clarify. Also, if you are going to leave these lists, either incorporate it into the paragraph as normal sentences, or make it more clear that you are deliberately including a list (e.g. with bullet points).
Comment: Changes have been made. We agree with you regarding readability, which is why we used bullet points in the original version, but they were removed during editing in the editorial office. We don't want to enter another table, that's why key markers are listed in the full sentence.
Page 8, first full paragraph: Not sure what you mean by “Exponents of active inflammation…” Do you mean markers of active inflammation?
Comment: The word " Exponents" has been replaced with " Mediators and effectors".
Single Nucleotide Polymorphisms
Page 8, just prior to table: The sentence “more than fifty SNPs…” is a nearly exact copy from the Wang et al. article that is cited. This is plagiarism and needs to be rewritten.
Comment: We are sorry for the mistake, the sentence was too close to Wang et al.. We paraphrased it and now it: “For example, Wang et al. [117] in their overview and meta-analysis described 56 SNPs from different genes that are associated with either hip OA (eg. in COL11A1, TGF-1β , VEGF), knee OA (eg. IL-6, ASPN, GDF5), or both joints (eg. IL-8, CALM1, SMAD3)”.
Table 2: You cite your main reference for this table as the Wang et al meta-analysis. However, not all of the SNPs listed in this table are found in this reference (for example, those in MMP3 and TIMP3, neither of which are mentioned in the Wang paper). If additional resources were used, then they should be cited appropriately. Further, it is incorrect to state that Wang identified these SNPs – in fact, this group summarized the findings from 7 other studies that were determined to meet the inclusion criteria for their meta-analysis. Additionally, their meta-analysis did not uphold the associations reported in the original studies for nearly all of these SNPs. When reporting results from genetic studies, it is crucial to differentiate between statistical association and mechanistic results. You have not provided that context in this section.
Comment: The table has been changed. Now is:
|
Gene |
SNPs |
Affected joint and impact |
Reference |
|
MMP-3 |
rs639752, rs520540, rs602128, rs650108, rs679620 |
· increased risk of hip, knee, hand OA · breakdown of extra-cellular matrix proteins |
[76, 117] |
|
MMP-8 |
rs1940475, rs3765620 |
· common allelic variants associate with knee and hand OA predisposition (conflicting or inconclusive) |
[118] |
|
TIMP-3 |
rs715572 (G/A), rs1962223 (G/C) |
· association with hip, knee OA risk in the recessive model · rs715572G/A could be a candidate protective gene for severe knee OA · interactions between Smad3 rs6494629 T/C and TIMP3 rs715572 G/A polymorphisms coul have protective roles in knee OA |
[117, 119] |
|
COL11A1 |
rs1241164, rs4907986, rs2615977 |
· association with hip OA risk |
[120] |
|
COL9A2 |
rs7533552 |
· asociation with hip OA · G allele in COL9A2 changes a glutamine to arginine or to tryptophan (changes a polar and thus important change in load-bearing cartilages stabilizing the collagen fibrils) |
[121] |
|
VEGF |
rs833058 |
· association with hip OA in men |
[120] |
|
DVWA |
rs7639618, rs9864422, rs11718863 |
· risk factor for developmental dysplasia of the hip and knee etiology (conflicting or inconclusive) |
[122-124] |
|
FRZB |
rs7775, rs288326 |
· risk factor for hip, knee and hand OA (conflicting or inconclusive) · relationship between proximal femur shape and incident radiographic hip OA (rs288326) |
[125, 126] |
|
GDF5 |
rs143383 (risk allele T) |
· risk factor for hip, knee and hand OA (conflicting or inconclusive) |
[126, 127] |
|
CALM1 |
rs12885713 (-16C/T transition SNP) |
· association with the pathogenesis of hip and knee OA (conflicting or inconclusive, probably dependent on populations) · −16T allele decreases CALM1 transcription in vitro and in vivo · correlation with articular chondrogenesis regulation and adhesion of chondrocytes to extracellular matrix proteins during the cartilage repairing process |
[128-131] |
|
TGF-1β |
rs1982073, rs1800470 (TT genotype), rs180046 (TT genotype and T allele), rs1800469 (TT genotype and T allele) |
· increase the individual susceptible of OA · contribution in disturbances of homeostasis and cartilage formation |
[119, 132] |
|
IGF1 |
rs2195239, rs11247361 (4488C>G) |
· increased risk of knee, hip, hand and spine OA · osteoarthritic cartilage degradation |
[133-136] |
|
IL1RN |
rs9005, rs315952, rs419598, rs315943, rs315920 (C/t) |
· risk factor for hip, knee and hand OA (conflicting or inconclusive) · marker of progression of knee and hip OA |
[137-142] |
|
IL-6 |
rs1800795 (−174G>C) |
· association with hip OA (conflicting or inconclusive) |
[135, 143] |
|
MCF2L |
rs11842874 (risk allele G) |
· risk factor for knee OA · association with reduction in pain and improvement in function for knee OA patients |
[144-146] |
|
SMAD3 |
rs12901499 (GA and GG genotypes), rs12102171 (CC and CT+TT and TT and TG+GG), rs2289263, rs6494629 (T/C), |
· together with BMI may be susceptible risk factors to OA · interactions between SMAD3 rs6494629 T/C and TIMP3 rs715572 G/A polymorphisms coul have protective roles in knee OA |
[119, 147, 148] |
|
BMP5 |
rs921126 (GA and GG genotypes) |
· significantly increased risk of knee OA |
[148] |
|
ADAM12 |
rs1871054, rs3740199 (Gly48Arg) |
· association with female knee OA (conflicting or inconclusive) · candidate for the prevalence and progression of knee OA |
[135, 149] |
|
ADAMTS14 |
rs4747096 |
· association with knee OA and Achilles tendon pathology |
[150-153] |
Epigenetic Changes
Page 10: The reorganization of this section is somewhat confusing. You have moved part of a paragraph about miRNA up, but then follow it by a mixed paragraph talking about methylation and miRNA, and then follow that by a long and very detailed section on specific miRNAs again.
Comment: The first 2 paragraphs in this section deal with introductory information about possible epigenetic changes. The second paragraph provides more detailed information on miRNAs because they are post-transcriptional expression regulators with particular and high therapeutic potential. The last paragraph, similarly to the previous ones, contains a discussion of this group of biomarkers on specific examples. We have combined information on methylation and miRNA when they relate to the same gene (IL-1β expression level regulation). This is a thoughtful organization of the section and we do not want to change the concept.
Gene Therapy Approaches
Marrow stimulation
Page 11, bottom paragraph: The first sentence (“The effectiveness…”) does not make grammatical sense and I’m not sure what it is referring to. You don’t actually explain what you mean by marrow stimulation – are you talking about micropicking, or something else?
Comment: Marrow stimulation is the term used to describe all the surgical techniques using bone marrow to enhance the articular cartilage defect healing. In this part, we presented to gene therapy approaches. The first one using genetically modified bone marrow and the second one where the viral vector is used directly to the exudate that enters the osteochondral lesion. So the first sentence was changed to simplify the meaning of marrow stimulation into “There are two gene-based simple techniques of marrow stimulation.”
ACI
The end of this section feels incomplete. You state that a cohort of cells was transduced, but don’t state what the outcome was. Are you simply stating that it can be done? Please clarify.
Comment: We have added additional information about the outcome according to Your instruction: “There were no treatment-related serious adverse events. Although knee evaluation scores seemed to indicate a trend toward efficacy, patient numbers were not sufficient to determine statistical significance.”
Muscle and Fat Grafts
In contrast to the rest of this section, you do not provide concrete examples of gene therapy for this section. Without this information, it doesn’t really fall under “Gene Therapy Approaches”.
Comment: In this section we presented the result of the study by Evans et. al (Evans, C. H.; Liu, F. J.; Glatt, V.; Hoyland, J. A.; Kirker-Head, C.; Walsh, A.; Betz, O.; Wells, J. W.; Betz, V.; Porter, R. M.; Saad, F. A.; Gerstenfeld, L. C.; Einhorn, T. A.; Harris, M. B.; Vrahas, M. S., Use of genetically modified muscle and fat grafts to repair defects in bone and cartilage. Eur Cell Mater 2009, 18, 96-111.) According to your suggestion we have changed the text to provide concrete information about this approach into ‘’Results from pilot experiments with rabbits showed that using adenovirus vectors carrying BMP2 cDNA is encouraging. The implanted genetically modified fat and muscle grafts formed bone in the subchondral region and cartilage above, indicating the impact of the progenitor cell location on the process of differentiation.’’
Scaffolds
Figure 3: I cannot see that there is a difference between C and D in the new figure. The description in the text of the figure legend describes the process as well or better than the figure itself. Unless changed significantly, this figure is unnecessary.
Comment: The figure was changed according to Your instructions from the previous review. We added the D to show that the viral vectors spread into surrounding tissue after the implantation. We think that the figure could supplement the information described in the text.
Page 13, paragraph above the table: The edited first sentence no longer makes sense as written. The sentence at the end of this paragraph can also be significantly shortened for clarity: “Table 3 presents different scaffold-based approaches that have been reported.”
Comment: We agree with Your comment. We have changed the first sentence into: “Among the numerous synthetic materials, biocompatible and biodegradable compounds matrices are thought to possess the most promising potential for repair of osteochondral defects.”

Round 3
Reviewer 1 Report
IJMS-832403 (2nd Revision) Genetics in cartilage lesions: from basic science to clinical practice
Overview: This is a review article that presents some of the known information about gene expression in cartilage and synovium as well as potential opportunities for gene therapy in the treatment of osteochondral disease.
General Comments: The formatting changes made this revision much easier to read and are much appreciated. Overall, most concerns have been addressed. The occasional syntax error or odd word choice remains (a few examples are below), but this can be corrected with careful copyediting. The biggest concern remaining is that you are not always clear about the species of the various animal models that you present. See specific comments below.
Line 164: I am unsure what you mean by “a decrease in the growth of MMP-3 and ADAMTS-5”. Do you mean that they did not increase as much (or as rapidly) as untreated, or that they were decreased compared to untreated? Please clarify.
Line 181: Should read “Growth factors are…”
Line 261-262: The sentence beginning “For this reason…” is not needed and can be removed.
Lines 280-282: This sentence doesn’t quite make sense as written. Could be simplified to: “…independently identified a missense mutation (rs143383) in the GDF5 promoter.”
Line 306: “minimally-invasive” would be preferable to “low-invasive”
Lines 323-324: What was injected in this study? Need to define what you mean by “the agomir group” – I assume that this is something that was injected, but you need to make this clear. Also, very important to state what species this experiment was conducted in.
Lines 329-337: Here and elsewhere, please make sure that you are identifying at least the species (if not the specific model) for these studies. This provides important context for assessing the results you are reporting.
Line 355: The word “simple” should be removed from this sentence, as it implies a judgement of the ease of the process. If you mean that they are “straightforward” (i.e. not many steps), then use this word instead (and it should go before the phrase “gene-based”).
Line 436: Replace “were placed” with “are shown”. There are several examples of minor syntax or grammar errors in this paragraph.
Line 446-447: I’m not sure what you mean by “…can be one of the existing cell-based cartilage repair methods.” Either it is, or it isn’t. If you are talking about this from a clinical perspective, genetically modified MSCs are definitely not being used, although regular MSCs are (as you state in the next sentence). This paragraph is an example of why it is important to explicitly state species and/or specific model being used for the results you present, because you give an example of human clinical practice (MSCs) followed immediately by two examples of experimental studies done in other species – for someone not familiar with the field, they may assume you are still talking about things done in people.
Line 480: You throw in a statement here about delivering genes to the synovium, but I can’t find where you introduce this concept in the main text above. If the concept is not discussed earlier, it shouldn’t be thrown into the conclusion in this way.
Author Response
July 24, 2020
IJMS-832403 (2nd Revision) Genetics in cartilage lesions: from basic science to clinical practice
Dear Editor and Reviewer
Thank You for giving us the opportunity to submit a revised draft of our manuscript. We truly appreciate You and the Reviewer time and effort in providing feedback on our manuscript. We have been able to incorporate the requested changes within the manuscript which have been highlighted throughout. Below is a point-by-point response to the Reviewer’s feedback.
We hope you agree that the comments raised by the Reviewer have been appropriatelyaddressed, allowing the revised manuscript to be suitable for publication in the International Journal of Molecular Sciences.
Sincerely,
Joanna Szczepanek
On behalf of all co-authors
Overview: This is a review article that presents some of the known information about gene expression in cartilage and synovium as well as potential opportunities for gene therapy in the treatment of osteochondral disease.
General Comments: The formatting changes made this revision much easier to read and are much appreciated. Overall, most concerns have been addressed. The occasional syntax error or odd word choice remains (a few examples are below), but this can be corrected with careful copyediting. The biggest concern remaining is that you are not always clear about the species of the various animal models that you present. See specific comments below.
Line 164: I am unsure what you mean by “a decrease in the growth of MMP-3 and ADAMTS-5”. Do you mean that they did not increase as much (or as rapidly) as untreated, or that they were decreased compared to untreated? Please clarify.
Response: We agree with Your suggestions. The sentence: “Researchers showed that passive motion loading suppresses immobilization-induced overexpression of the MMP-3 and ADAMTS-5 genes, as well as blocking the increase in total articular cartilage MMP-3 activity.”, has been added for clarification.
Line 181: Should read “Growth factors are…”
Response: The change has been made.
Line 261-262: The sentence beginning “For this reason…” is not needed and can be removed.
Response: The change has been made.
Lines 280-282: This sentence doesn’t quite make sense as written. Could be simplified to: “…independently identified a missense mutation (rs143383) in the GDF5 promoter.”
Response: The change has been made.
Line 306: “minimally-invasive” would be preferable to “low-invasive”
Response: The change has been made.
Lines 323-324: What was injected in this study? Need to define what you mean by “the agomir group” – I assume that this is something that was injected, but you need to make this clear. Also, very important to state what species this experiment was conducted in.
Response: According to Your suggestion, we have clarified the description of the presented study by Si et al. Currently, the description is: "An example of research into the therapeutic role of miRNA is the study by Si et al. [83], which explored the possibility of slowing the progression of induced surgically OA in model rats using intra-articular injections of a miRNA-140 mimic or inhibitor. MiRNA-140 is specifically expressed in cartilage, and its role includes regulation of ECM-degrading enzymes. Overexpression of miRNA-140 in primary human chondrocytes has been shown to promote collagen II expression and inhibit the expression of target genes: MMP-13 and ADAMTS-5. Si et al. [83] observed, that in the group treated with miRNA-140 agomir, significantly higher results were obtained in cartilage thickness, chondrocyte count and collagen II expression level, while simultaneously reducing expression levels of MMP-13 and ADAMTS-5. Thus, the potential of miRNA-140 in attenuation OA progression by modulating ECM homeostasis was demonstrated. "
Lines 329-337: Here and elsewhere, please make sure that you are identifying at least the species (if not the specific model) for these studies. This provides important context for assessing the results you are reporting.
Response: According to Your suggestion, we have clarified the description of the presented study by Wang et al. and Huang et al. Currently, the description is: “Similar observations were obtained by Wang et al. [182] for miR-483-5p (MMP-2 expression regulator) in transgenic (TG483) mice model. Researchers, using the appropriate constructs introduced intraarticularly, could both enhance degenerative disease (by introducing LV3-miR-483-5p lentivirus) and delay the progression of experimental OA (using antago-miR-483-5p) [182]. Interesting results have been described by Huang et al. [181], who conducted intraarticular injection of recombinant AAV for in vivo overexpression of miR-204 in surgically induced OA mice model.”
Line 355: The word “simple” should be removed from this sentence, as it implies a judgement of the ease of the process. If you mean that they are “straightforward” (i.e. not many steps), then use this word instead (and it should go before the phrase “gene-based”).
Response: We agree with your suggestion. The word ‘’simple’’ was removed from this sentence.
Line 436: Replace “were placed” with “are shown”. There are several examples of minor syntax or grammar errors in this paragraph.
Response: We replaced ‘’were placed” with “are shown” according to Your suggestion. The paragraph was corrected by the native speaker.
Line 446-447: I’m not sure what you mean by “…can be one of the existing cell-based cartilage repair methods.” Either it is, or it isn’t. If you are talking about this from a clinical perspective, genetically modified MSCs are definitely not being used, although regular MSCs are (as you state in the next sentence). This paragraph is an example of why it is important to explicitly state species and/or specific model being used for the results you present, because you give an example of human clinical practice (MSCs) followed immediately by two examples of experimental studies done in other species – for someone not familiar with the field, they may assume you are still talking about things done in people.
Response: We agree with Your suggestions. The sentence: “Genetically modified MSCs can be one of the existing cell-based cartilage repair methods” was placed after an example of human clinical practice (MSCs) and changed into: ‘’Although genetically modified MSCs are not being used in human clinical practice, there were done some experimental studies.” We also added information about the specific model used in the presented studies. We removed the sentence “In another study, Venkatesan et al. designed 3D fibrin-polyurethane scaffolds in a hydrodynamic environment that provided a favorable growth environment for rAAV-infected Sox9- modified hBMCs and promoted their differentiation into chondrocytes [226].” To the next paragraph not to mix animal studies with those done on human MSCs.
Line 480: You throw in a statement here about delivering genes to the synovium, but I can’t find where you introduce this concept in the main text above. If the concept is not discussed earlier, it shouldn’t be thrown into the conclusion in this way.
Response: We agree with Your suggestion. We removed the statement about delivering genes to the synovium.